



# Black Carbon physical and optical properties across northern India during pre-monsoon and monsoon seasons

James Brooks[1], Dantong Liu[2], James D. Allan[1,3], Paul I. Williams[1,3], Jim Haywood[4,5], Ellie J. Highwood[6], Sobhan K. Kompalli[7], S. Suresh Babu[7], Sreedharan K. Satheesh[8], Andrew G. Turner[3,6], Hugh Coe[1].

[1] Centre for Atmospheric Science, School of Earth and Environmental Sciences, University of Manchester, Manchester, UK.

[2] Department of Atmospheric Sciences, School of Earth Sciences, Zhejiang University, Hangzhou, Zhejiang, China.

[3] National Centre for Atmospheric Science, Manchester, UK.

[4] Observation Based Research, Met Office, Exeter, UK.

10   [5] College of Engineering, Mathematics & Physical Sciences, Exeter, UK.

[6] Department of Meteorology, University of Reading, Reading, UK.

[7] Space Physics Laboratory, Vikram Sarabhai Space Centre, Thiruvananthapuram, Kerala, India.

[8] Centre for Atmospheric & Oceanic Sciences, Indian Institute of Science, Bengaluru, India.

*Correspondence to*: Prof. Hugh Coe (hugh.coe@manchester.ac.uk)



**Abstract.**

Black carbon (BC) is known to have major impacts on both climate and human health, so is therefore of global importance, particularly so in regions close to large populations that have strong sources. The physical properties and mixing state of black carbon containing particles are important determinants in these effects but information is often lacking, particularly in some of the most important regions of the globe. Detailed analysis into the vertical and horizontal BC optical and physical properties across northern India has been carried out using airborne in-situ measurements. The size-resolved mixing state of BC-containing particles was characterised using a single particle soot photometer (SP2). The study focusses on the Indo-Gangetic Plain during the pre-monsoon and monsoon seasons. Data presented are from the UK Facility for Airborne Atmospheric Measurements BAe-146 research aircraft that performed flights during the pre-monsoon (11[th] and 12[th] June) and monsoon (30[th] June to 11[th] July) seasons of 2016.

Over the Indo-Gangetic Plain, BC mass concentrations were greater (1.95 µg/m$^3$) compared to north-west India (1.50 µg/m$^3$) and north-east India (0.70 µg/m$^3$) during the pre-monsoon. Across northern India, two distinct BC modes were recorded; a mode of small BC particles (core diameter < 0.16 µm and coating thickness < 50 nm) and a mode of moderately-coated BC (core diameter < 0.22 µm and coating thickness 50-200 nm). The Indo-Gangetic Plain and north-east India locations exhibited moderately-coated black carbon particles with enhanced coating thicknesses, core sizes, mass absorption cross sections and scattering enhancement values compared to much lower values present in the north-west. The coating thickness and mass absorption cross section increased with altitude (13%) compared to the boundary layer. As the monsoon arrived across the region, mass concentration of BC decreased over the central Indo-Gangetic Plain and north-east locations (38% and 28% respectively), except for the north-west location where BC properties remained relatively consistent. Post-monsoon onset, the coating thickness, core size, mass absorption cross section and scattering enhancement values were all greatest over the central Indo-Gangetic Plain much like the pre-monsoon but were considerably reduced over both north-east and north-west India. Increases in mass absorption cross section through the atmospheric column were still present during the monsoon for the north-west and central Indo-Gangetic Plain locations, but less so over the north-east due to lack of long-range transport aerosol aloft. Across the Indo-Gangetic Plain and north-east India during the pre-monsoon and monsoon, solid fuel (wood burning) emissions form the greatest proportion of BC with moderately-coated particles. However, as the monsoon develops in the north-east there was a switch to small uncoated BC particles indicative of traffic emissions, but the solid fuel emissions remained in the IGP into the monsoon. For both seasons in the north-west, traffic emissions form the greatest proportion of BC particles.

Our findings will prove important for greater understanding of the BC physical and optical properties, with important consequences on the atmospheric radiative forcing of BC-containing particles. It will also be useful for BC source and emission inventory studies in the future.



# 1 Introduction

Increased anthropogenic emissions into the atmosphere, especially over south Asia, has led to severe air quality issues. Understanding, identifying and characterising air pollution sources and their effect on local and regional areas, is important for health impact studies as well as radiative forcing assessments on local and global scales (Lawrence et al., 2007). The Indo-Gangetic Plain (IGP) in northern India is one such polluted region in south Asia (Pawar et al., 2015; Singh et al., 2017). Aerosol mass sources within the IGP are dominated by natural sources such as dust, but also anthropogenic emissions associated with traffic, biomass burning, waste incineration and crop and wood burning (Banerjee et al., 2015; Singh et al., 2017). Black Carbon (BC) is one of the crucial components of ambient aerosol released into the atmosphere due to the incomplete combustion of fossil fuels, bio-fuels and biomass burning (Bansal et al., 2019). Due to its strong absorption at a wide range of wavelengths, BC is considered to be a prime contributor to radiative heating of the atmosphere (Marinoni et al., 2010). Strong light absorption from BC coupled with its longer atmospheric lifetime makes BC an important aerosol constituent affecting regional climate (Bansal et al., 2019).

India is currently the second-largest BC emitter in the world, with emissions projected to rise steadily in the coming decades (Rana et al., 2019). Unlike the USA, UK and Europe where aerosol BC is predominantly sourced from on-road and off-road diesel engines (Bond et al., 2013) with a shift towards solid fuel burning in winter (Liu et al., 2014), Indian BC emissions are overwhelmingly from low-efficiency combustion of domestic fuels (47%) followed by industrial emissions (22%) (Paliwal et al., 2016). Regional BC emission inventories have considerable variability owing to the emission factors, unreliability of fuel consumption estimates and the general lack of detailed in-situ measurements (Bond et al., 2013; Rana et al., 2019). Additional uncertainties are sourced from variable physio-chemical and optical properties of BC aerosol (Koch et al., 2009; Lee et al., 2013). The aging and internal mixing of BC aerosol in the atmosphere with components such as sulphate, organic carbon and secondary organic aerosol can enhance BC light absorption by 30-100% as compared to an external mixture (Liu et al., 2017; Rana et al., 2019). Therefore, the BC mixing properties are of importance for further study and clarification across India, especially northern India where BC mass concentrations are known to be significant (Bond et al., 2013; Brooks et al., 2019).

Environmental regulation targeted emissions from fossil fuel combustion by transport, power plants and a variety of industrial activities. However, residential solid fuel burning for space heating purposes (such as coal or wood burning) has received less attention in terms of enforcement or regulation during the past decade (Liu et al., 2014). Especially for Europe, there has been a growing number of studies that highlight the importance of residential burning and their contribution to particulate matter (PM) loadings, with emphasis on winter time studies when domestic heating activities are high and boundary layer mixing is suppressed (Herich et al., 2011; Liu et al., 2011; Laborde et al., 2013; Crippa et al., 2014). Organic mass fractions in submicron aerosols contributed by solid fuel sources reported in these studies ranged from 15-50% consistently using receptor-based source apportionment methodologies, highlighting the importance of wood burning as a source contribution (Herich et al., 2014). The methodologies outlined above have mostly focused on the organic carbon content of



solid fuel sources. However, as Liu et al (2014) explains the source information for black carbon aerosols (BC) is sparse in part due to the difficulties of measuring the refractory component in the aerosols. Given BC is the principal source of particulate light absorption in the atmosphere (Bond et al., 2013) with adverse effects on human health (e.g. Jansen et al., 2005; Mordukhovich et al., 2009), it is of great importance to understand BC sources and how its properties are dictated by those

sources.

Prior to our study, only a few surface based measurements of BC using an SP2 had been made (Raatikainen et al., 2017; Thamban et al., 2017; Kompalli et al., 2019). The data presented in this work represent the first vertically resolved measurements of BC mass and mixing state using an SP2 during the pre-monsoon and monsoon seasons over northern India. Previous studies (Rana et al., 2019) have been unable to identify particular sources of BC from ambient measurements, though

some insight into larger BC particles has been possible (Kumar et al., 2011; Vadrevu et al., 2012; Kaskaoutis et al., 2014). Previously SP2 data from urban environments such as London and China have been used to determine contributions to BC from different sources based on the core size and coating thickness, the latter being represented as scattering enhancement (Liu et al., 2014; 2019). Over London during winter, two distinct BC sources were observed, one of small BC core diameters with low scattering enhancement that was identified as being from traffic emissions, and the second of larger BC core diameters

with increased scattering enhancement (solid fuel-type emissions). These sources could be clearly separated since the measurements were made in close proximity and the BC types were distinctly different (Liu et al., 2014; 2019).

In view of the large variations associated with BC emission inventories in the India region, model outputs of BC mass and single site location studies need to be validated against high-temporal and spatial airborne atmospheric measurements (Rana et al., 2019). This paper presents the first airborne measurements of BC physical properties over northern India using

the SP2 instrument. It is the first time that BC physical properties are measured quantitatively at such temporal and spatial scales, providing detailed insights on source-specific BC properties during the pre-monsoon and monsoon.

## 2 Methodology

Ten science flights form this study, that were conducted by the UK Facility for Airborne Atmospheric Measurement (FAAM) BAe-146 research aircraft with the flight tracks highlighted in Figure 1 and flight summaries in Table 1. The flights

took place during two periods: the pre-monsoon (11th and 12th June 2016) and the monsoon onset period (30th June to 11th July 2016), based at Lucknow (LKN; 26.85°N, 80.95°E). The aircraft flew with a comprehensive instrument suite, capable of measuring aerosols, cloud physics, chemical tracers, radiative fluxes and meteorological fields, however only instruments used in this analysis are discussed further. The FAAM BAe-146 has a typical range of ~3000 km and an altitude ceiling of over 10 km, with an aircraft science speed of ~100 ms$^{-1}$. From the operating base, the aircraft typically covered radial distances of

~200-300 km in 4.5/5 hours of flight time, resulting in over 120 hours flying completed throughout the campaign (89 hours science). Based on the likely synoptic and local conditions on the day, different types of science flights were conducted; radiation flights and survey flights. Both flight types consisted of long-leg duration flights covering the NE Bay of Bengal area





(BBA) and Indo-Gangetic Plain (IGP) regions, delivering the main part of the aerosol characterisation. In addition, profiles to high altitudes when taking off from Lucknow and in selected other locations were carried out in order to build up a statistical picture of the vertical structure. Low altitude straight-level runs (SLRs) were also carried out at heights of around 0.5-1.0 km.

## 2.1 Instrumentation

The physical properties of individual refractory BC particles (rBC, Petzold et al., 2013) were characterised using a Single Particle Soot Photometer (SP2) manufactured by Droplet Measurement Technologies (DMT) Inc. (Boulder, CO, USA). The SP2 data was corrected to standard temperature and pressure (STP) of 273.15 K and 1013.25 hPa respectively. The instrument operation and data interpretation procedures are covered in detail elsewhere (Liu et al., 2010; McMeeking et al., 2010). Briefly, the SP2 uses an intra-cavity Nd:YAG laser at 1064nm to determine the optical size of a single particle by light
scattering and, if material within the particle absorbs at the laser wavelength, the refractory mass of the particle is quantified by detection of the laser induced incandescence radiation. In the atmosphere the main light-absorbing component at this wavelength is BC (Liu et al., 2014). The SP2 incandescence signal was to obtain single particle rBC mass after calibration using Aquadag sample black carbon particle standards (Aqueous Deflocculated Acheson Graphite, manufactured by Acheson Inc., USA), with a correction value of 0.75 required to fully represent ambient particles (Moteki and Kondo, 2010; Laborde et
al., 2012; Baumgardner et al., 2012). The measured rBC mass is converted to a mass equivalent diameter (1.8 gcm$^{-3}$ for atmospheric BC (Bond and Bergstrom, 2006)), which is termed the BC core diameter ($D_c$), which is the diameter of a sphere containing the same mass of rBC as measured in the particles. The scattering signal of a BC particle will be distorted during its transit through the laser beam because of the mass loss of a BC particle by laser heating, thus the leading-edge scattering signal before the onset of volatilisation is extrapolated to reconstruct the scattering signal of a BC-containing particle (Gao et
al., 2007).

The physical properties, such as the coating thickness for a given single BC particle, is obtained by using an inverse Mie scattering model in conjunction with the BC core size. As described by Taylor et al (2014), this technique will obtain the equivalent diameter of a sphere with the BC assumed to be a concentric spherical inclusion with the same scattering cross section as the measured particle after Leading Edge Only (LEO) fitting. The technical details for the method of leading edge
only (LEO) fitting can be found in Liu et al (2014). The optical diameter of a BC particle or the coated BC size ($D_p$) is derived by inputting the LEO fitted scattering signal and BC core size into Mie calculations, and using a core refractive index (m) =2.26–1.26i (Moteki et al., 2010) and a coating refractive index m =1.5+0i. The relative or absolute coating thickness of a BC particle is calculated as $D_p/D_c$ and $(D_p - D_c)/2$, respectively, with the absolute coating thickness presented in this study. The optical size of a non-BC particle is again calculated using Mie theory using m = 1.5 + 0i, thus the optical sizes of coated BC
and non-BC are directly comparable using the analysis here. Given the coating thickness for individual particles is $D_c$ size dependent, a bulk coating thickness is evaluated as the cubed root of the total volume of the BC particles divided by the total volume of BC cores, as expressed in Equation 1:



$$\frac{D_p}{D_c} = \left(\frac{\sum_i D_{p,i}^3}{\sum_i D_{c,i}^3}\right)^{\frac{1}{3}} \qquad (1)$$

where $D_p$ and $D_c$ are the coated BC diameter and BC core diameters respectively; $i$ denotes the $i$th single BC particle. The volume weighted bulk $D_p/D_c$ is considered to be a representative diagnostic for the overall mixing state of the entire population of BC particles.

To best illustrate the BC core and coating methodology, a parameter of scattering enhancement, $E_{sca}$, can be introduced, which is defined by Equation 2:

$$E_{sca} = \frac{S_{coated}}{S_{uncoated}^*} \qquad (2)$$

where $S_{coated}$ is the scattering signal measured from the SP2 and then LEO fitted. $S_{uncoated}^*$ is the scattering signal of the corresponding BC core, with the asterisk denoting it is calculated using the Mie single particle scattering solutions. Following

this methodology, a value of $E_{sca} = 1$ refers to a BC particle that scatters equivalently to that containing only a BC core, in other words the BC particle has zero coating ($D_p/D_c = 1$). However, as Liu et al (2014) explains, particles with any associated coatings will scatter more than the core, thus $E_{sca}$ will be > 1; $S_{coated}$ will also be subject to instrument measurement uncertainty, therefore a fraction of particles of $E_s < 1$ would be expected. An increase of $E_{sca}$ will coincide with a thicker coating thickness for a specified $D_c$.

The volume-weighted coated BC size ($D_p$) size distribution is calculated as the product of the bulk relative coating thickness and the MMD of the BC cores, to indicate the mean coated BC size (Equation 3):

$$D_{p,v} = \frac{D_p}{D_c} \; x \; MMD \qquad (3)$$

The mass absorption coefficient (MAC) at λ=880nm to avoid the influence of brown carbon (BrC) absorption at shorter wavelengths is calculated for each single particle by assuming the refractive index of rBC core 1.95±0.79i (Bond and

Bergstrom, 2006; Laborde et al, 2013) and coating thickness refractive index 1.50±0i (Liu et al., 2015), using the Mie core-shell approach (Bohren and Huffman, 2008). The MAC in bulk for a given time is calculated as the integrated absorption coefficient (MAC x $m_{rBC}$) for all particles divided by the integrated particle masses, expressed by Equation 4:

$$MAC = \frac{\sum_i MAC_i \; x \; m_{rBC,i}}{\sum_i m_{rBC,i}} \qquad (4)$$

Where $MAC_i$ and $m_{rBCi}$ are the MAC and rBC mass for each single particle respectively. The calculation is performed for each

type of BC. The MAC is an important property for BC particles as it has important ramifications for radiative transfer simulations. The MAC is size dependent, resulting in larger values for particles with thicker coatings or associated water at high relative humidity (Schnaiter et al., 2005). Instrument limitations have in the past led to large discrepancies in MAC values throughout the literature (Bond and Bergstrom, 2006), however the SP2 measures the refractory BC mass independently of the mixing state (Laborde et al., 2013). Our analysis has removed the influence of dust on the BC mass concentration.



## 3    Results

The pre-monsoon (11[th] – 12[th] June, flights of B956 and B957) and monsoon (30[th] June – 11[th] July, flights of B968-B976) are considered separately. Of this second category, the first few flights can be regarded as occurring during the monsoon transition phase, since the monsoon has arrived at some locations but not others. For example, during B968 (30[th] June) the monsoon was seen to be influencing the IGP but not Jaipur in NW India. More information on the monsoon development can be seen in appendix Figure A1 (in this example for 30[th] June, the monsoon progression isochrone for 2016 still lies to the south and east of Jaipur), Figure A2 for mean pre-monsoon and monsoon wind directions in and above the boundary layer, and in Brooks et al (2019). The analysis presented below first outlines the BC characteristics within the boundary layer from the Straight Level Runs (SLRs) measurements during the pre-monsoon and monsoon transition period across northern India, followed by the vertical profile analysis (section 3.1). Further analysis presents the semi-direct climatic impact potential of the BC across northern India, highlighted the various properties and scattering enhancement potential the BC exhibits (section 3.2) along with the BC mass absorption cross section (MAC; section 3.3).

As highlighted in Brooks et al (2019), the aerosol burden over northern India has a distinct structure in the pre-monsoon. An elevated aerosol layer is present between 3-6 km, particularly over northwest India with somewhat decreased extent in the far northeast of India. The aerosol chemical composition remained largely similar as the monsoon season progressed, but the total aerosol mass concentrations decreased by ~50% as the rainfall arrived; the pre-monsoon average total mass concentration was 30 µg/m³ compared to a monsoon average total mass concentration of 10-20 µg/m³. However, this mass concentration decrease was less noteworthy (~20-30%) over the Indo-Gangetic Plain, likely due to the strength of emission sources in this region. In the aerosol vertical profile, inside the Indo-Gangetic Plain during the pre-monsoon, organic aerosol and absorbing aerosol species dominated in the lower atmosphere (<1.5 km) with sulphate, dust and other scattering aerosol species enhanced in an elevated aerosol layer above 1.5 km with maximum aerosol height ~6 km. As the monsoon progressed into this region, the elevated aerosol layer diminished, the aerosol maximum height reduced to ~2 km and the total mass concentrations decreased by ~50%.

## 3.1  BC physical properties

### 3.1.1 Boundary layer BC

Summary statistics of boundary layer aerosol chemical composition for northern India can be found in figures 2, and more detailed analysis can be found in figures 4. The boundary layer was predominantly above 1500m, such that the Straight Level Run (SLR) data presented are within the boundary layer throughout.

Greatest black carbon mass concentrations were present inside the IGP (average 1.95 µg/m3), with decreased mass concentrations outside in both the NW (1.5 µg/m³) and NE of India (0.7 µg/m³). The BC core MMD was larger inside the IGP (0.25 µm) compared to the NW (0.22 µm) and NE (0.22 µm) of India. The coating thickness of the BC particles shows some similarity between the NW and IGP regions, with a slightly larger average outside the IGP in the NW (1.7) compared to inside




the IGP (1.65). Over NE India, the coating thickness was significantly larger potentially due to the prevailing wind direction carrying larger coated BC particles to the region, with an average of 1.9 witnessed.

Moving into the monsoon onset period, there were changes seen but also similarities in the BC physical properties across northern India. Black carbon mass concentrations decreased somewhat across all regions but remained relatively
consistent compared to the pre-monsoon. Over the IGP, mass concentrations decreased by ~25-40% as the monsoon progresses across the region, with average mass concentrations of ~1.2 µg/m³. In NW India, mass concentrations fluctuated significantly between flights but overall no significant decrease in average concentrations occurred. This is potentially due to the NW region witnessing consistent wind directions from the west from Pakistan and beyond, therefore bringing similar air masses from the same source to the region. In the NE region of India, the BC mass concentrations were lower than over the IGP, with decreases
witnessed as the monsoon progressed. Concentrations were somewhat elevated over locations near to the built-up region of Bhubaneswar (0.50 µg/m³) and close to the IGP boundary during B975 (0.50 µg/m³) compared to mass concentrations of ~0.25 µg/m³ in between.

With monsoon progression came increases in the coating thicknesses of particles across northern India, especially over the IGP. Increases of up to 35% were witnessed in the absolute coating thicknesses of BC particles from ~1.6 to ~2. An
increase in absolute coating thicknesses was also witnessed over NW India, albeit smaller than the increases over the IGP, with coating thickness averages increasing from ~1.7 to ~1.8. Over NE India close to the IGP boundary, coating thicknesses were similar to the central IGP coating thickness values (~2) whereas over Bhubaneswar the coating thicknesses were much smaller ~1.6 during the monsoon flights. This could be due to the wind directions between the locations, with NE India influenced by local sources as the wind originates mostly from the Bay of Bengal, whereas closer to the IGP the air mass had
greater influence from the larger, solid fuel type sources of the IGP.

A similar pattern existed regarding the core MMD regional characteristics. Outside the IGP in the NW, core mass median diameters were relatively constant, with averages ~0.2 µm throughout the monsoon progression period, again likely due to consistent wind direction from west of the region. Over the IGP however, no significant changes were seen in core size, with averages in the PM at 0.25 µm compared to 0.25 µm during the pre-monsoon progression. Over the NE of India, core
MMD values were quite consistent despite monsoon progression with averages ~ 0.22 µm during the PM and ~0.20 µm in the monsoon period.

### 3.1.2 Vertical distribution of BC

Vertical profiles were also carried out using the aircraft in NW India (Jaipur (26.91ºN 75.79ºE)/Jodhpur (26.24ºN 73.02ºE)), the IGP (Lucknow (26.85ºN 80.95ºE)) and NE India (Bhubaneswar (20.30ºN 85.83ºE)) during both the pre-
monsoon and monsoon transition periods, as shown in the summary plot Figure 3 and in Figure 5. The boundary layer (BL) height during the pre-monsoon is chosen as 2 km and during the monsoon is 1.5 km, following the explanation in Brooks et al (2019).



Throughout the vertical profile, distinct structure can be seen in the black carbon mass concentration and the black carbon physical properties. In the pre-monsoon in NW India, black carbon mass concentrations were consistent through the profile with averages ~0.25 µg/m³. Black carbon core MMD also showed consistent values through the aerosol profile in NW India, with averages of ~0.22 µm. Black carbon coating thickness however displayed increasing values with altitude, with

values ~1.60 in the boundary layer compared to ~1.9 aloft at 3km. Inside the IGP, the black carbon mass concentration in the EAL was of similar order to NW India with an average of 0.50 µg/m³, with core MMD average of 0.25 µm and coating thickness average of 1.85. A similar pattern occured over NE India at Bhubaneswar, with black carbon mass concentrations of ~0.6 µg/m³ in the boundary layer, with similar concentrations aloft. Aloft however, there was a clear EAL highlighted in the black carbon core MMD and coating thickness information. In the BL, core MMD was ~0.19 µm and coating thickness was

1.80. In the EAL increases were witnessed in both parameters with core MMD ~0.21 µm and coating thickness of 1.90.

Progressing into the monsoon onset period, variations were witnessed in the mass concentrations and the physical properties of BC. In the NW region, few changes are seen to the BC properties and mass due to the late monsoon arrival there. Mass concentrations were consistent at pre-monsoon levels ~0.25 µg/m³ with core MMD averages similar also at 0.22 µm. Coating thickness however did show some variations, with values greater in the boundary layer ~2 and ~1.82 aloft from 2-4

km.

Changes were witnessed over the IGP as the monsoon progressed over the region. BC mass concentrations decreased most significantly in the EAL. The maximum aerosol height was reduced from ~7 km during the pre-monsoon to ~1.5-2 km during the monsoon, with mass concentrations decreasing from ~0.5 µg/m³ in the EAL to negligible concentrations during the monsoon. In the boundary layer, decreases also occurred but not as significantly as aloft, with averages ~2 µg/m³ in the early

monsoon (B970/B972) onset to ~1 µg/m³ as the monsoon is more developed over the region (B973/B974). The BC core MMD values remained consistent despite monsoon arrival, with averages of 0.20 µm. From B970 to B973, coating thickness increased with altitude (~1.55 in BL compared to ~1.9 aloft between 1.5-3 km), but overall decreased throughout the profile as the monsoon progressed. The coating thickness did however increase during B974 compared to the previous flights (~1.6 in BL and ~2.05 aloft). Similar changes were seen over BBA in the NE region as the monsoon progressed. The maximum

aerosol height decreased from ~5 km during the pre-monsoon to ~2.5 km during the monsoon season, with mass concentrations decreasing in the EAL from ~0.6 µg/m³ to negligible concentrations due to EAL removal. In the boundary layer, mass concentrations were witnessed also from ~0.6 µg/m³ in the pre-monsoon to ~0.3 µg/m³ during the monsoon season. BC coating thickness still displayed some increases with altitude during the monsoon season, but of less significance compared to the pre-monsoon. The core MMD values decreased with altitude during the monsoon season, with values ~0.20 µm in the BL to ~0.17

µm aloft between 1.5-3 km.

## 3.2 BC scattering enhancement and size distributions

The variation of scattering enhancement ($E_{sca}$) as a function of BC core diameter ($D_c$) is shown in figures 6 and 7, with BC core diameter ($D_c$) and coated diameter ($D_p$) size distributions in figures 8 and 9 respectively.



During the pre-monsoon, the NW location showed small BC particles with very little coating (<50 nm). The IGP and NE India however show a greater proportion of the BC particles with a greater scattering enhancement, with an increase in core diameter and coating thickness (~100 nm). The wind directions in Appendix Figure A2 can shed some light on these features. During the pre-monsoon, the IGP air mass is transported eastwards carrying the BC generated in the region into the
north-east of India. As a result, there is a similar distribution of scattering enhancement observed across both locations. For the NW India location however, the wind direction is from the west, receiving air from areas such as the Thar Desert. The BC core diameter ($D_c$) size distribution and coated BC ($D_p$) histogram provide consistency with other findings in our study. During the pre-monsoon, the BC number and mass BC core size distributions are significantly enhanced in the IGP region compared with the NW and NE regions which are lower in number and mass. Across the three regions, the peak BC core diameter was
similar at ~182 nm. The coated diameter ($D_p$) histogram was consistent with the $E_{sca}$ information, with the BC containing particles over NE India being larger and more coated (0.21 µm) compared to NW India (0.15 µm).

During the monsoon, the NW region was influenced by the monsoon progression resulting in a reduction in the number of BC particles and to a lesser extent in the scattering enhancement, though small BC particles with very little coating (<50 nm) were again dominant as in the pre-monsoon. The distribution of $E_{sca}$ for black carbon particles in the IGP region
during the monsoon was similar to that of the pre-monsoon but with much reduced particle numbers indicative of monsoon washout and change of wind direction (see Appendix Figure A2). Increased scattering enhancements compared to the NW, with moderately-coated BC particles (~100 nm) were observed similar to the pre-monsoon. The scattering enhancement values were much different in the north-east with the monsoon arrival. The prevailing wind direction in the region was from the marine environment in the Bay of Bengal with some influence from air masses travelling from middle India and there was
little influence from the IGP region. This is reflected in the scattering enhancements, with much lower quantities of moderately-coated BC particles, with the environment of Bhubaneswar dominated by small BC particles with small absolute coating thicknesses (<50 nm). The BC core diameter ($D_c$) size distribution and coated BC ($D_p$) histogram present changes as the monsoon developed in the regions sampled. For NE India during the monsoon, the BC core diameter mass size distributions are consistent across B971 and B975 (182nm respectively), with the IGP BC core MMD larger at 210 nm. This is similar to
the other IGP BC core diameter mass size distributions for the monsoon transition and monsoon flights, due in part to the consistent wind direction. During B975 compared to B971, however, decreases were seen in the BC mass concentration and core diameter size distribution during the monsoon period, a feature also observed in the BC core diameter number concentrations. The BC coated diameter ($D_p$) histogram is consistent with this, with the peak in the distribution occurring at a diameter of 0.25 µm in the IGP compared to 0.12 µm for both monsoon flights over NE India. This highlights the stark
differences in BC properties between NE India and the central IGP locations.

We have shown that increased scattering enhancements are present over the IGP and NE India during the pre-monsoon, compared to lower scattering enhancements over all locations during the monsoon season. Instances of high (low) scattering enhancement coincide with large (small) MAC and CT values. We have presented similar sources in our dataset to





previous research (Liu et al, 2014; 2019), but with clear increased mixing between sources due to the high amounts of secondary aerosol formation and photochemical aging across northern India.

In the north-west region, the BC properties highlight the greatest proportion of particles are small in core diameter and thinly coated. These types of BC particles have been shown to be traffic-dominated emissions in previous research (Liu et al, 2014). The north-east of India BC in the monsoon season see a large proportion from such traffic emissions due to the prevailing wind direction from the Bay of Bengal, meaning that BC particles witnessed are likely to be from the urban environment around Bhubaneswar where the aerosol was sampled. Previous research has highlighted that traffic emissions can form a large proportion of BC particles in east coast regions (Ramachandran, 2005). In the IGP during the pre-monsoon and monsoon, and north-east India during the pre-monsoon, moderately-coated BC particles are present in our analysis. This BC source can be appointed to solid fuel (wood burning) emissions, as explained in Liu et al (2014; 2019). Despite monsoon arrival the IGP BC characteristics remained similar despite some BC removal, with moderately-coated BC particles prevailing. This is consistent with the fuel use patterns (including forest fires/biomass burning episodes) across the IGP at this time of year (Kumar et al, 2011; Vadrevu et al, 2012; Kaskaoutis et al, 2014).

A source of BC that is not presented in the $E_{sca}$ plots across northern India is the large core size, thinly coated BC particles (BC core size >180 nm and coating thickness <50nm). Work by Liu et al (2019) in China explain that these BC particles have large core diameters with low scattering enhancements and are indicative of coal burning emissions. These do not appear to be present across northern India during the pre-monsoon and monsoon seasons in significant quantities. A reason could be that the BC particles from coal combustion are different in characteristics to the China coal emissions. Coal burning characteristics may vary between countries, so more detailed in-situ analysis would be required to understand more regarding Indian coal burning. Liu et al (2019) does explain that this type of BC particle may be uniquely present in urban Beijing, as it is not present in the UK or surrounding areas.

### 3.3 Mass absorption coefficient

The mass absorption coefficient (MAC) observed for the different regions across northern India are shown in Figure 2. During the pre-monsoon, the three locations of the NW, central IGP and NE India show distinct MAC properties. In the NW boundary layer, MAC averaged 7.39 m²g⁻¹ (σ 0.36 m²g⁻¹) compared to 6.76 m²g⁻¹ (σ 0.47 m²g⁻¹) inside the IGP and 8.08 m²g⁻¹ (σ 0.33 m²g⁻¹) over NE India. The MAC was lower in the IGP compared to other locations potentially due to the close proximity to the aerosol sources, with the MAC value high in the NW due to the long-distance transport of IGP aerosol to the NW in the prevailing pre-monsoon flow. A distinct vertical structure in the MAC was observed during the PM. Throughout all locations sampled in northern India, greater MAC values were observed aloft (~7.5 to 8.5 m²g⁻¹) compared to the BL (~6.5 to 7.5 m²g⁻¹; see Figure 4).

As the monsoon developed, changes in the MAC were evident for some but not all locations. For the NW region, BC MAC was consistent to pre-monsoon values at 7.44 m²g⁻¹ (σ 0.23 m²g⁻¹) likely due to the lack of full monsoon development in this region and consistency in emission sources, as stated in Brooks et al (2019). Whereas, the central IGP and NE regions



showed significant changes as the monsoon progressed. The MAC in the central IGP increases by 15% with an average of 7.97 m$^2$g$^{-1}$ (σ 0.29 m$^2$g$^{-1}$) in the monsoon compared to the pre-monsoon values, whereas NE India witnessed decreases in MAC by 17% to 6.91 m$^2$g$^{-1}$ (σ 0.38 m$^2$g$^{-1}$). During the monsoon the vertical profiles of MAC in the NW India locations and the IGP increased aloft compared to the BL. However, over the NE there was clear removal of aerosol aloft so BC mass concentrations, and therefore MAC values are only retrieved inside the BL (see Figure 5).

## 4    Discussions

During the pre-monsoon, Kompalli et al (2014) showed that strong thermal convection increases the boundary layer height compared to other seasons, and coupled with high wind speed, results in highly dispersed aerosol particles in both the vertical and horizontal directions reducing near-surface concentrations. Our observations show that the BC mass concentrations are less than half over NE India as they are in the central IGP during the pre-monsoon (0.70 and 1.95 µg/m$^3$ respectively), a spatial pattern consistent with Thamban et al (2017). The IGP BC mass concentrations are elevated somewhat compared to other locations across northern India due to the strength of the emissions sources as well as being in close proximity to the local emissions (Brooks et al., 2019). Our BC mass concentrations are comparable to mean values reported for urban regions, by Kompalli et al (2019) in north-east India (~0.8 µg/m$^3$), Liu et al (2014) in London (~1.3 µg/m$^3$) and greater than values in the city of Paris (~0.9 µg/m$^3$) in Laborde et al (2013). However, they are much lower than those reported in Chinese cities, such as Beijing (~5.5 µg/m$^3$; Wu et al., 2016) and Shenzhen (~4.1 µg/m$^3$; Huang et al., 2012) though the earlier studies are from urban surface sites and not regional aircraft measurements.

Our study provides new information on aerosol absorption over NE India. Previous work by Vaishya et al (2018) observed moderately high SSA values (0.8), decreasing with altitude indicating increased aerosol absorption in the lower free troposphere. BC physical and optical properties are important in the quantification of the aerosol semi-direct effect, with large coating thicknesses and the associated absorption enhancement being particularly implicated (Bond et al., 2013). Larger, thickly coated BC particles with large MAC values over the IGP could have an increased warming potential, affecting the climate impact of BC over northern India, particularly if present at elevated altitudes. Our observations show that thickly coated BC particles are present at high altitudes in the vertical profiles over the IGP and NE India in the pre-monsoon (see Figure 5) and are therefore likely to exert a strong warming effect. We observe that the BC layer is associated with an increase in coating thickness, core size and MAC over NE India during the pre-monsoon coincident with the larger particles observed by Vaishya et al (2018). Taken together these observations suggest that solid fuel burning is the likely source of these BC particles that are subsequently transported long distances from the IGP across to the NE India region. Vaishya et al (2018) showed that the absorption over Bhubaneswar in the pre-monsoon was enhanced. However, since their measurements were carried out using an Aethalometer they were unable to account for coating thickness or lensing effects. Liu et al (2014) showed that enhanced BC coating thickness and increased lensing effects resulted in increased absorption. Our measurements therefore



highlight that the absorption occurring over BBA could be of even greater magnitude than those presented by Vaishya et al (2018).

The scattering enhancement ($E_{sca}$) results are consistent with the BC optical properties presented, and aid in reinforcing the BC sources outlined. The BC mass absorption coefficient (MAC) is one such optical property that presents consistencies with the $E_{sca}$ and coating thickness. Laborde et al (2013) explain that MAC is often <7.5 when under strong influence from urban emissions and >8 for aged air masses. During periods where $E_{sca}$ presents strong traffic emission sources, such as in the north-west during pre-monsoon and monsoon and the north-east during monsoon, the $E_{sca}$ values suggest an urban influence due to the small coating thicknesses on the small core diameter BC particles, consistent with MAC values <7.5. Inside the IGP for the pre-monsoon and monsoon seasons, $E_{sca}$ analysis suggests solid fuel sources of BC with the moderately coated particles and large MAC values. During the shift in air mass direction over the north-east as the monsoon progresses, the BC optical properties change as the $E_{sca}$ undergoes changes with the aerosol aloft removed. Rather than an air mass arriving over north-east India from the north-west travelling through the IGP, south-easterly winds from the marine background of the Bay of Bengal prevailed, therefore the aerosol measured over north-east India was likely to be from local urban emission sources. This could explain the reduction in MAC from aged air mass values (>8) to more urban emissions (<7.5), the reduction in core diameter and reduced coating thickness, all indicative of urban traffic emissions (Laborde et al., 2013). The BC optical properties also prove useful for identifying the solid fuel burning BC particles over the IGP, consistent with previous literature where wood burning stoves, open fires and agricultural residue-based stoves are known to arise greatly across the IGP (Banerjee et al., 2015; Paliwal et al., 2016; Singh et al., 2017; Fleming et al., 2018). Solid fuel, as outlined by Liu et al (2014), is characteristic of larger core sizes, thicker coating thicknesses and larger MAC values. All these features listed are present in the IGP, as shown in Figure 2 and 3, for the pre-monsoon and monsoon seasons producing the large proportion of solid fuel BC particles.

## 5 Conclusions

The observations of the physical properties of atmospheric black carbon in India presented in this work are the first of their kind over the Indian subcontinent but the results are consistent and build-upon previous understanding regarding BC properties over northern India. An aircraft campaign was conducted from 11$^{th}$ June to 11$^{th}$ July 2016 to characterise black carbon optical and physical properties, during both the pre-monsoon and monsoon seasons. The study represented the north-west, central IGP and the north-east of India across the two seasons. Exhaustive measurements of the black carbon mass concentration, mixing state and source properties were carried out. As we move from outside to inside the IGP, the BC mass concentrations increase from 0.70 µg/m$^3$ in the north-east and 1.50 µg/m$^3$ in the north-west, to 1.95 µg/m$^3$ in the IGP region. As the monsoon progressed over northern India, BC mass concentration decreased over the IGP and north-east India (by 38% and 28% respectively), except for north-west India where mass concentrations remained relatively consistent.



BC aerosol across the IGP presented moderately-coated particles with enhanced coating thickness, core size and mass absorption coefficient, indicative of solid fuel burning particles. These source particles are witnessed over the north-east region due to the prevailing wind direction during the pre-monsoon. In the north-west however, the BC particles are small with very little coating and low mass absorption coefficients, suggestive of traffic emissions. As the monsoon progressed over northern

India, alterations in the atmospheric BC properties were witnessed. Over the north-east region, BC underwent changes to much smaller particles with very little coating mainly due to a switch in wind direction from the Bay of Bengal, resulting in very few solid fuel particles being transported from the IGP as in the pre-monsoon. The IGP again presented moderately-coated BC particles showing solid fuel sources into the monsoon season due to strong emission sources, with the north-west dominated by small BC particles from traffic sources consistent with the pre-monsoon environment. Vertical structure was found in the

BC properties with the coating thickness and mass absorption cross section increasing by 13% with altitude compared to the boundary layer across northern India, during the pre-monsoon and monsoon seasons. Relating this to previous research highlights the large absorption potential of the BC aerosol over northern India, especially aloft.

This BC source information, coupled with the extensive in-situ measurements, will prove pivotal in improving understanding behind potential radiative forcing from BC-containing particles over India and emission inventory work.

**Competing interests**

The authors declare that they have no conflict of interest.

**Author contributions**

JB was responsible for the SP2 instrument operation in the field, data processing, data analysis, and the writing of this paper. HC contributed to the writing of the paper. JA and PW were responsible for the maintenance and running of the AMS prior to

and during the campaign. DL supplied expertise and operation of the SP2 prior to and during the campaign and assisted with the data analysis. SK also contributed to the operation of the SP2 in the field. JH, EH, SB, SS, AT and HC were the project investigators for this campaign.

**5 Acknowledgements**

We would like to thank those involved in the SWAAMI project, which is part of the larger MONSOON project. This includes

the Facility for Airborne Atmospheric Measurements (FAAM) and AirTask who manage and operate the BAe-146 Atmospheric Research Aircraft, which is jointly funded by the Natural Environmental Research Council (NERC) and the Met Office. A number of institutions were involved in logistics, planning and support of the MONSOON campaign: The Met Office, University of Reading, Vikram Sarabhai Space Centre India, and the Indian Institute of Science India. ERA-Interim



wind field data was provided courtesy of ECMWF. The lead author was supported by a NERC studentship grant NE/L002469/1 and the authors supported by NERC grant numbers NE/L013886/1, NR/L01386X/1 and NE/P003117/1.



# Figures

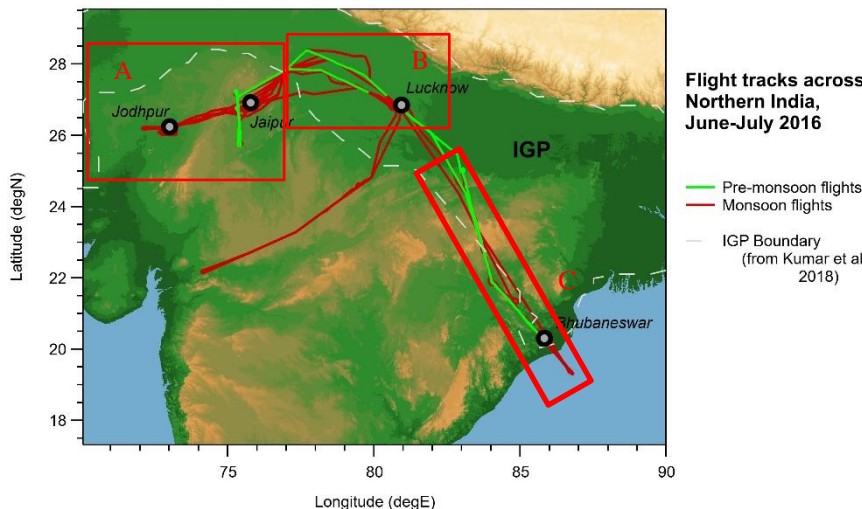

**Figure 1 Flight tracks of the BAe-146 aircraft for the campaign across India during the pre-monsoon and monsoon seasons of 2016. The flight paths considered by this analysis are described in the main text and Table 1. Straight Level Run (SLR) boundary layer sections are split by region (A) West of IGP in NW India, (B) IGP, and (C) South-East of IGP in NE India.**

| Flight | Season | Date | Depart (Z) | Return (Z) | Duration (hh:mm) | Operating region |
|--------|--------|------|------------|------------|------------------|------------------|
| B956 | PM | 11/06 | 03:05 | 07:36 | 04:31 | W |
| B957 | PM | 12/06 | 05:30 | 09:26 | 03:56 | E |
| B968 | PM/M | 30/06 | 03:32 | 07:28 | 03:56 | W |
| B970 | PM/M | 03/07 | 04:46 | 08:42 | 03:56 | W |
| B971 | PM/M | 04/07 | 05:40 | 10:05 | 04:25 | E |
| B972 | M | 05/07 | 03:27 | 07:29 | 04:02 | W |
| B973 | M | 06/07 | 02:10 | 06:41 | 04:31 | W |
| B974 | M | 07/07 | 04:27 | 08:18 | 03:51 | W |
| B975 | M | 09/07 | 04:29 | 09:04 | 04:35 | E |
| B976 | M | 10/07 | 04:23 | 08:51 | 04:28 | W |

**Table 1 Flight summary for operations included in this study. All flights were conducted in Northern India in the pre-monsoon (PM) and monsoon (M) season (PM/M season refers to transition period of when the monsoon was arriving in Northern India). The dates of the flights are shown, with their respective region of study.**





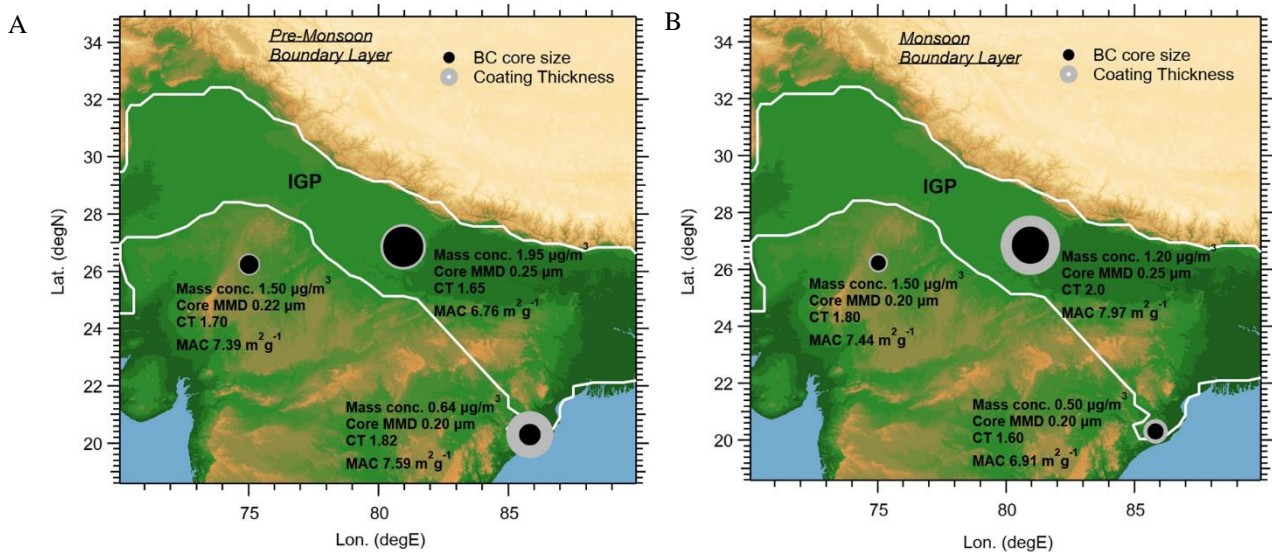

**Figure 2** Pre-monsoon (A) and Monsoon (B) average boundary layer (BL) black carbon optical and physical properties, across northern India. The BC core and coating information are presented for NW India between Jaipur and Jodhpur, the central IGP and NE India near Bhubaneswar. The IGP boundary (white line) is from Kumar et al (2018).

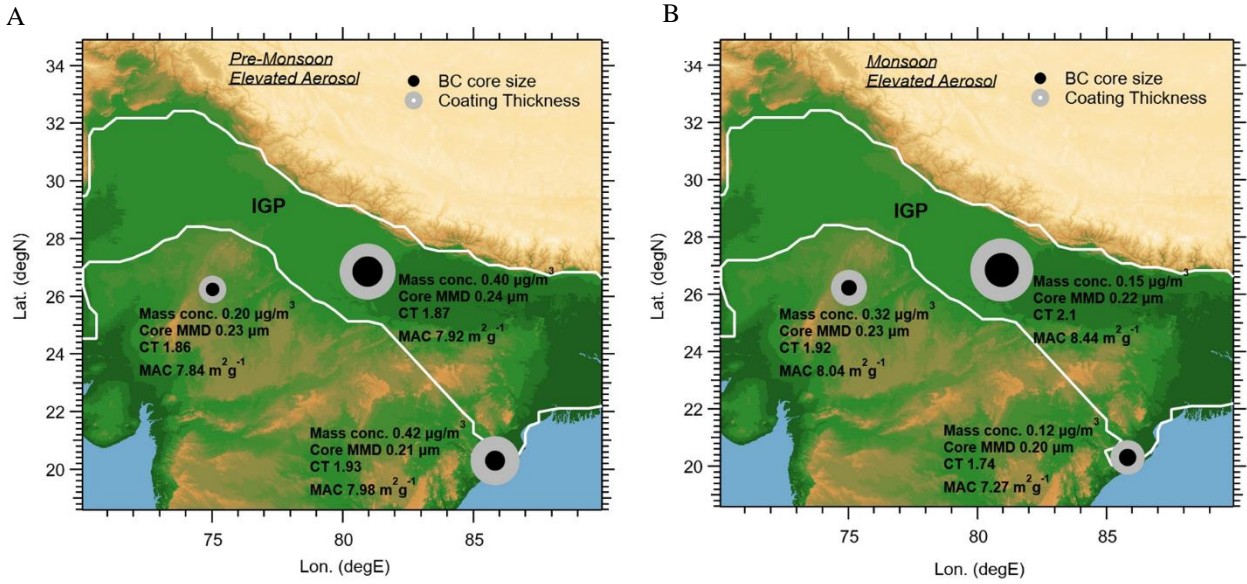

**Figure 3** Pre-monsoon (A) and Monsoon (B) average elevated aerosol above the boundary layer black carbon optical and physical properties, across northern India. The BC core and coating information are presented for NW India between Jaipur and Jodhpur, the central IGP and NE India near Bhubaneswar. The IGP boundary (white line) is from Kumar et al (2018).





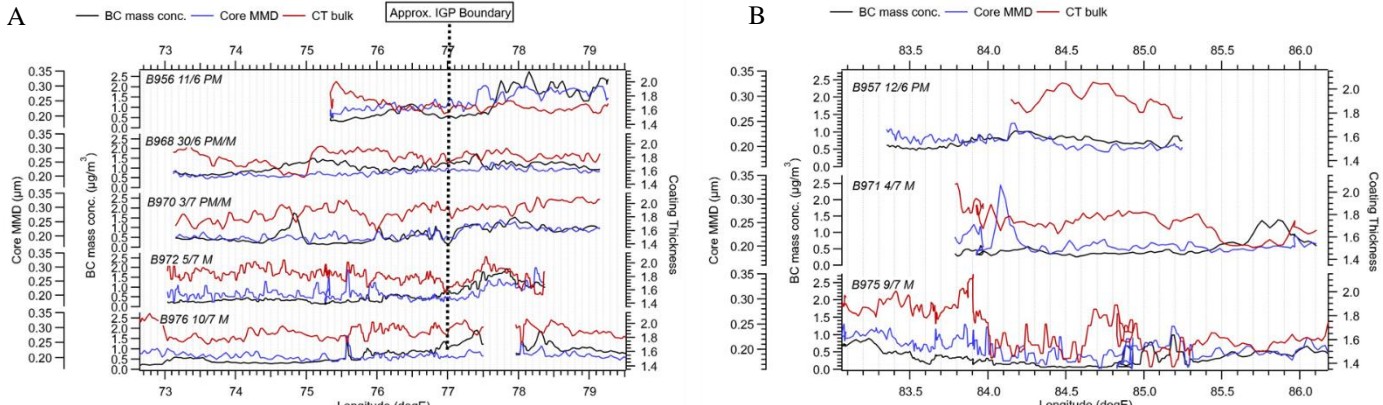

**Figure 4 Horizontal spatial BC physical properties for pre-monsoon (PM) and monsoon (M) seasons in northern India, from LKN to JAI/JOD (panel A) and from LKN to BBA (panel B). The plots highlight the IGP region (77°E-eastwards) and outside the IGP (73-77°E) separated by the vertical black dashed line. The monsoon progression is indicated by the blue shaded region.**

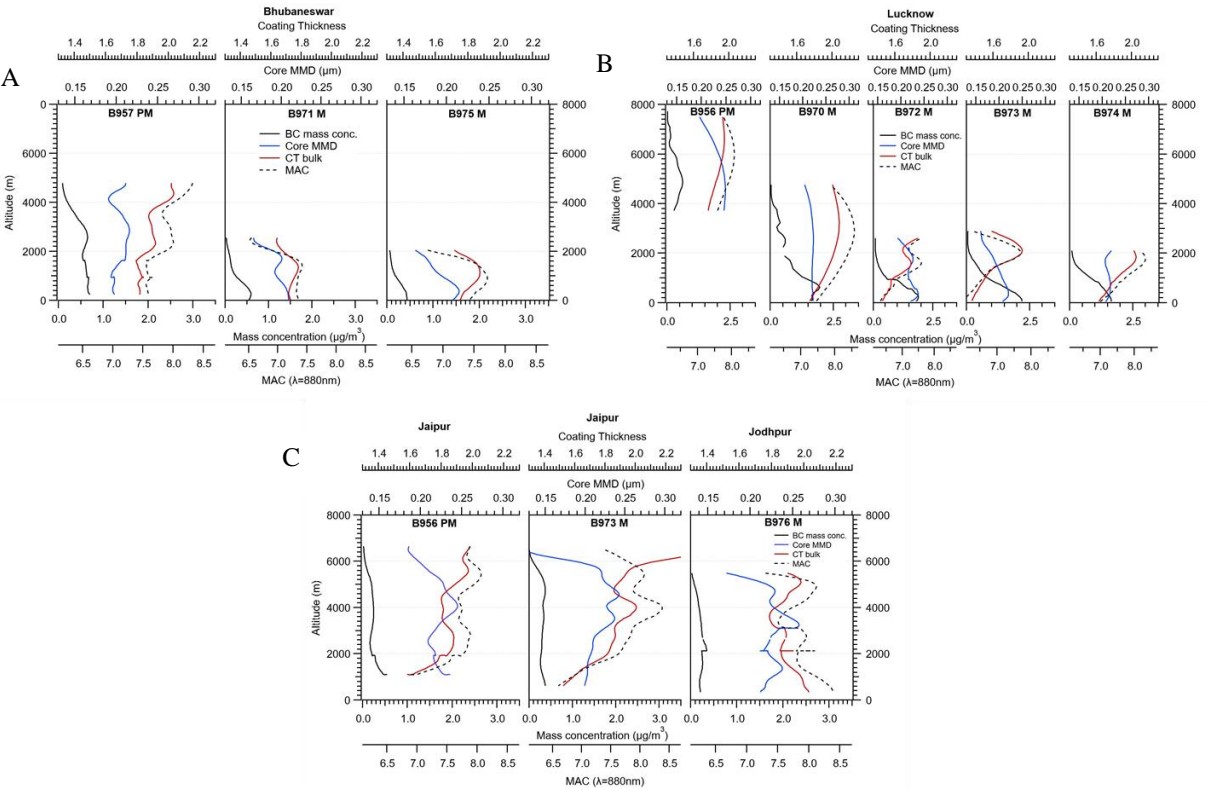

**Figure 5 Vertical profile BC physical properties for (A) Jaipur/Jodhpur in NW India, (B) Lucknow in the IGP, and (C) Bhubaneswar in NE India. Data presented are for the pre-monsoon (PM) and monsoon (M) seasons.**





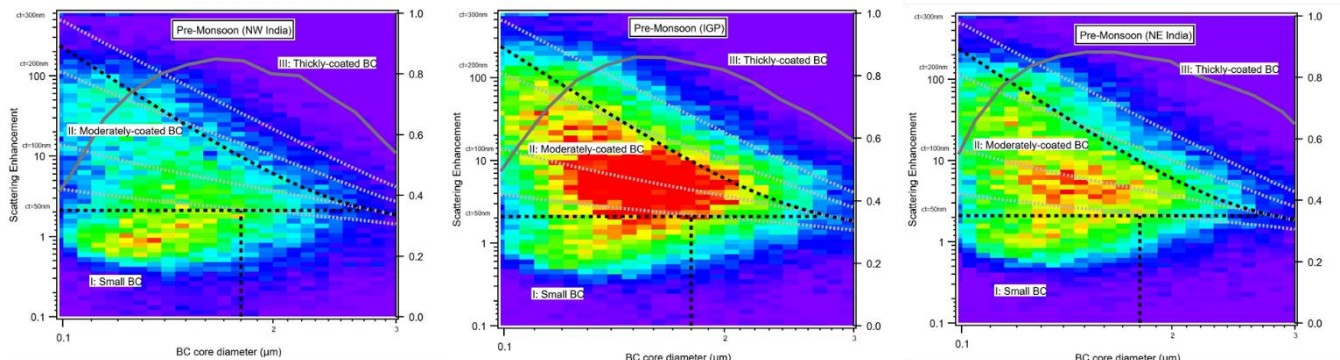

**Figure 6** BC optical properties the pre-monsoon (PM) season with the location sampled highlighted in the plot title. The plots present the scattering enhancement ($E_s$) as a function of BC core diameter ($D_c$). The image plot is a two dimensional histogram for the detected particles. The dashed grey contours show the absolute coating thickness (nm, ($D_P$-$D_c$)/2). The solid grey line, with corresponding scale on right axis, shows the number fraction of BC particles that were successfully determined according to their scattering signal at each $D_c$ size.

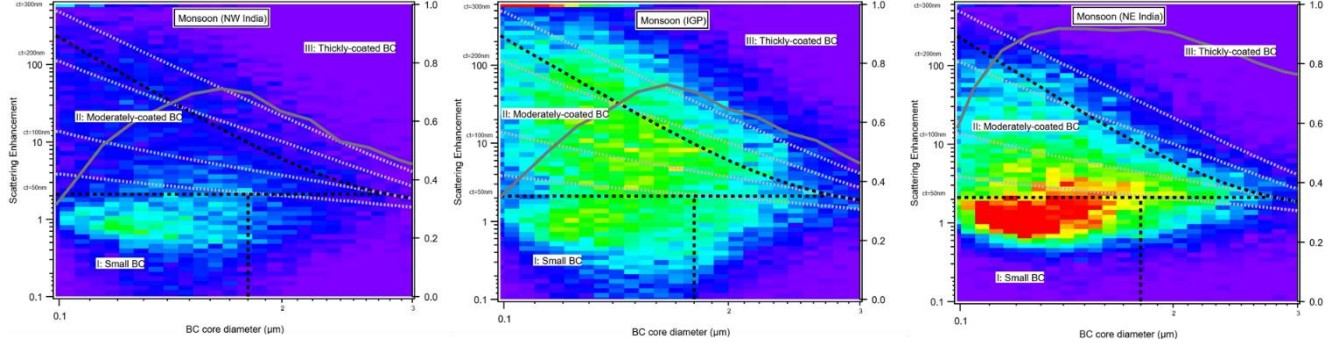

**Figure 7** BC optical properties for the monsoon (M) season with the location sampled highlighted in the plot title. The plots present the scattering enhancement ($E_s$) as a function of BC core diameter ($D_c$). The image plot is a two dimensional histogram for the detected particles. The dashed grey contours show the absolute coating thickness (nm, ($D_P$-$D_c$)/2). The solid grey line, with corresponding scale on right axis, shows the number fraction of BC particles that were successfully determined according to their scattering signal at each $D_c$ size.





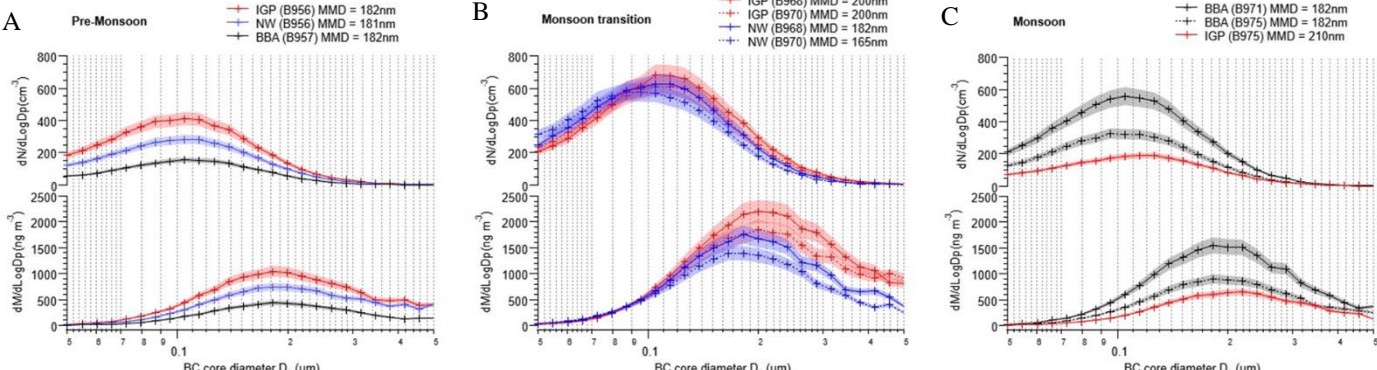

**Figure 8** BC core diameter ($D_c$) size distributions classified by (A) pre-monsoon, (B) monsoon transition and (C) monsoon season. The shading is the square root (%) errors for each distribution. Orange represents the IGP, with blue representing NW India (JAI/JOD) and black for NE India (BBA).

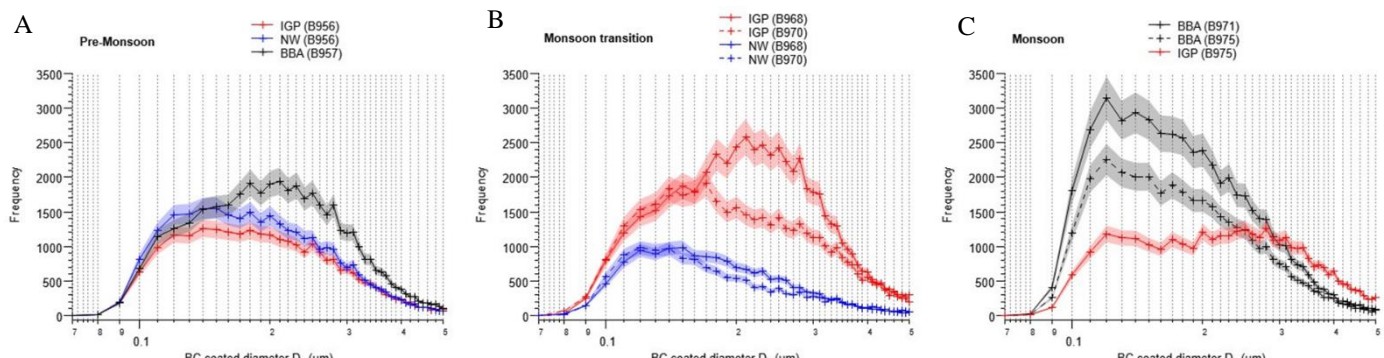

**Figure 9** BC coated diameter ($D_p$) histograms for (A) pre-monsoon, (B) monsoon transition and (C) monsoon season. The shading is the square root (%) errors for data series. Orange represents the IGP, with blue representing NW India (JAI/JOD) and black for NE India (BBA).



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





**Appendix**

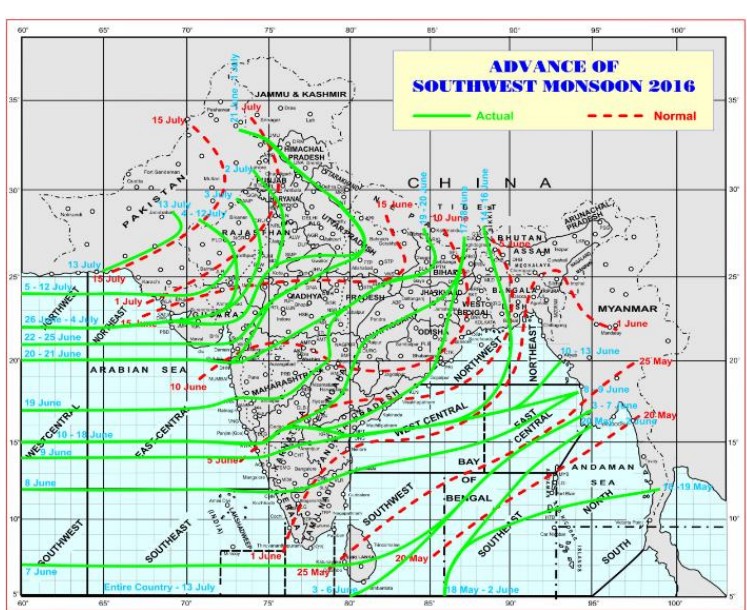

**Figure A1 Advance of the Indian summer monsoon of 2016 (India Meteorological Department, Ministry of Earth Sciences, India)**

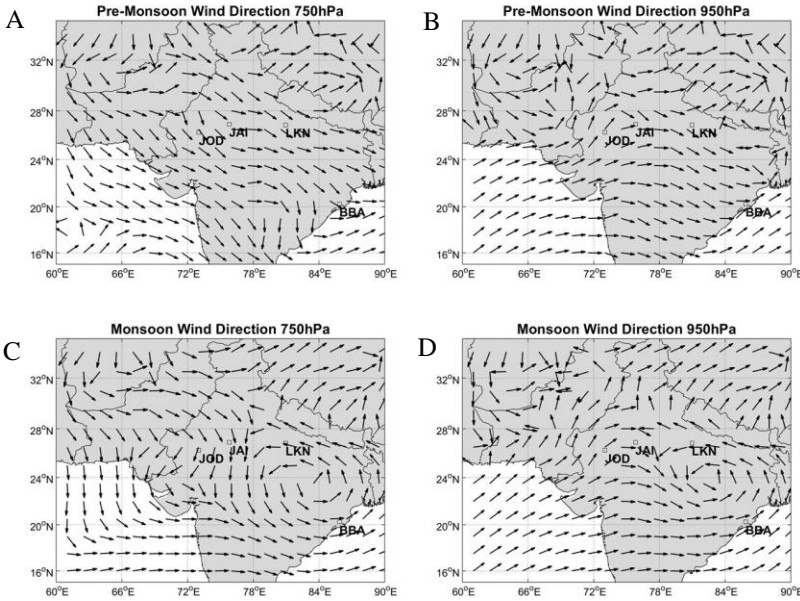

**Figure A2 Mean wind direction conditions during the pre-monsoon (panel A and B) and monsoon season (panel C and D). The data used in the maps is ERA-Interim (Dee et al., 2011).**