# Peer review of "Black Carbon physical and optical properties across northern India during pre-monsoon and monsoon seasons"

_Atmospheric Chemistry and Physics, 2019_

## Referee Comment (RC1) · Anonymous Referee #1 · 9 Jul 2019

The submitted manuscript presents detailed airborne in situ measurements of Black Carbon (BC) aerosols taken during different flights over northern India (Indo-Gangetic Plain or IGP) covering pre-monsoon and monsoon seasons. The characteristics of aerosols over the region are presented regarding high-quality vertical and spatial measurements of optical and microphysical properties of BC. The measurement dataset reveals higher concentration of BC over central IGP than that over northwest and northeast region during pre-monsoon season. Also, the BC particles over IGP and northeast regions are found to be moderately coated with enhanced core size and increased strength in mass absorption and scattering. Notably, these effects are seen pronounced at higher altitudes than in boundary layer. With the arrival of monsoon,

the concentration of BC particles reduced considerably over IGP and northeast, but remained relatively consistent in northwest. Larger coating thickness, core size of BC, and enhanced absorption and scattering strengths across Indo-Gangetic Plain and north-east India during the pre-monsoon and monsoon are indicative of solid fuel burning that forms the greatest proportion of BC over the study region.

The airborne measurements presented in the paper bring new detailed information about spatial and vertical distribution of BC aerosols over northern India that can help improve our understanding of role of BC in radiative forcing and constraint aerosol representation in the chemistry-climate models. It is practically hard to verify each and every observation reported in this paper. However, the findings are generally consistent with those reported in earlier studies and follow a general understanding of aerosol variability and meteorology in the region.

During the review, I came across two major points that author should consider while revising the paper. First, the authors present and discuss the coating of BC particles and associated optical and microphysical properties, but do not mention about the possible coating of BC on coarse mode dust particles. The latter situation is as important as the former one since it poses a great potential of changing the particles' radiative properties, thereby radiative forcing. Second, the paper doesn't discuss, at least briefly, the similarities and/or discrepancies between the measured BC properties and those assumed in the climate-chemistry models for the study region. A general discussion highlighting the importance of new measurements given the current understanding of BC aerosols would greatly enhance the scientific value of the paper.

The article is generally well-written, however, needs some attention to improve the presentation, e.g., long sentences, punctuations, and ambiguities. The content presented in the paper certainly fits into the scope of the ACP journal and can be published given above two major points are addressed. I am listing below some specific suggestions on the paper with this report.   Page 2, Abstract, line 17: "..compared to that in the boundary layer" Page 2, Abstract: Define Indo-Gangetic Plain as IGP first in the

abstract and use IGP for rest of the abstract to reduce the word count. Follow same terminology in Introduction and onwards. Page 2, Abstract, line 29-31: The findings will also help constraint the regional aerosol models for a variety of applications such as space-based remote sensing, chemistry transport model, and air quality. Page 3, Introduction, line 5-7: IGP serves as a unique natural aerosol laboratory influenced by seasonal sources of aerosols. Authors may cite specific studies/papers out there that highlight seasonal loadings of dust and anthropogenic aerosols. Page 3, Introduction, line 8-9: This is now a general understanding supported by numerous studies. Citing here the IPCC report or some high-impact factor on this subject would be more appropriate. Page 3, Introduction, line10: How about citing some benchmark papers of Tami Bond, who has done fundamental research on BC and its role in radiative heating. Page 3, Introduction, line13: I assume here that the central/southern Africa is the largest source of BC, and it is worth to mention it here with a corresponding citation. Page 3, Introduction, line 16-17: What is the contribution of agricultural biomass burning in BC loadings over IGP? The region undergoes intensive crop residue burning in post-monsoon (paddy burning) and pre-monsoon (wheat burning) emitting substantial amounts of carbonaceous aerosols into the atmosphere. Page 4, line 6: Define S2P here and then use abbreviation for the rest of the paper. Page 8, line 24: Some inconsistencies here. Please rewrite the sentence. Page 8, about coating thickness: Can the measurements and analysis shown here infer the properties of BC coating material? Mineral dust? Sulfate, Nitrate? Or humidification? Also, pre-monsoon period is also characterized with transported dust particles in the region. How about coating of dust with BC? Did the measurements/analysis show BC coating coarse mode particles? This is important because studies have shown that the absorption properties of BC coating over dust are significantly different (more absorbing, lower SSA) than pure mineral dust (less absorbing, higher SSA). Page 8, line 29: I wouldn't locate Bhubaneswar in NE; it is actually towards South-East of Indian subcontinent. Page 8, line 30: Shouldn't it be Figure 4 and Figure 5? Page 9, line 8-10: Confusing. Please consider restricting the sentences. Page 9: What is EAL? It isn't defined anywhere in

the manuscript. Page 11, Mass Absorption Coefficient, 1st paragraph: The possible reason for lower MAC over IGP and relative higher MAC in NW and NE isn't understood. Page 11, line 28-29: is it because of possibilities of elevated fine dust particles coated with BC?

---

## Referee Comment (RC2) · Anonymous Referee #2 · 12 Jul 2019

Reviewer Comments to Author(s):

This manuscript presents the physical and optical properties of black carbon (BC) over northern India using a single particle soot photometer (SP2). The study mainly focusses on the Indo-Gangetic Plain (IGP) during the pre-monsoon and monsoon seasons. The ms brings detailed information about spatial and vertical distribution of BC aerosols that can help to improve understanding of the role of BC in radiative forcing and climate models.

The work presented in the manuscript well fits into the scope of the journal. Overall, the article is well-structured; however, needs some attention to improve the value/quality

of the manuscript.

General comments: The authors need to discuss more details about the coating of black carbon, dust particles. How the current work is relevant and implements to climate models over the study area? Also, authors need to use more abbreviations like IGP, BC, SP2 throughout the manuscript.

I am listing below some specific suggestions that can help to improve the quality of the manuscript.

Abstract  c The abstract need to revise as the current description is very general, some information possible to move (line 2-10) in the introduction section.  c Use abbreviation of Indo-Gangetic Plain (IGP) as it repeated in several places (line 11, 14, 18, 21, 23, 24 and 27).  c Please use shorter sentences instead of a long sentence (line 19-24).

Introduction  c Page 3: line 2, need references to support the sentence  c Page 3: line 13-24, the second paragraph need to revise with good connection with earlier sentences. Also possible to discuss studies which are mainly focusing on BC source apportionment over South Asia.  c Page 4: line 19-21, possible to combine these two sentences

Methodology This section is too descriptive so better to move some information is supplement information, this will make more space for results and discussion.

Results  c Page 7, line 2, Please start this important section with topic sentences rather than measurements  c Page 7, line 18-19, use IGP abbreviation  c Page 7, line 15-18, Please explain this in more details  c Page 8, 1-2, Need to add more details to support this statement

Conclusions Please revise this full section with more specific outcomes.

Author contributions Please write a full description of SWAAMI project.

Figures Figure 1, 2 and 3: Use the different color font for text in the figure

---

## Author Comment (AC1) · 8 Sep 2019

We thank both reviewers for their constructive comments, which as outlined below have helped improve the manuscript. This document outlines the review comments in plain italics, followed by the authors replies in bold.

acp-2019-505-RC1 The submitted manuscript presents detailed airborne in situ measurements of Black Carbon (BC) aerosols taken during different flights over northern India (Indo-Gangetic Plain or IGP) covering pre-monsoon and monsoon seasons. The characteristics of aerosols over the region are presented regarding high-quality vertical and spatial measurements of optical and microphysical properties of BC. The mea-

surement dataset reveals higher concentration of BC over central IGP than that over northwest and northeast region during pre-monsoon season. Also, the BC particles over IGP and northeast regions are found to be moderately coated with enhanced core size and increased strength in mass absorption and scattering. Notably, these effects are seen pronounced at higher altitudes than in boundary layer. With the arrival of monsoon, the concentration of BC particles reduced considerably over IGP and northeast, but remained relatively consistent in northwest. Larger coating thickness, core size of BC, and enhanced absorption and scattering strengths across Indo-Gangetic Plain and north-east India during the pre-monsoon and monsoon are indicative of solid fuel burning that forms the greatest proportion of BC over the study region. The airborne measurements presented in the paper bring new detailed information about spatial and vertical distribution of BC aerosols over northern India that can help improve our understanding of role of BC in radiative forcing and constraint aerosol representation in the chemistry-climate models. It is practically hard to verify each and every observation reported in this paper. However, the findings are generally consistent with those reported in earlier studies and follow a general understanding of aerosol variability and meteorology in the region.

We thank the Reviewer for their encouraging view of our work and the useful suggestions for consideration, which we have incorporated in the revised manuscript.

During the review, I came across two major points that author should consider while revising the paper. First, the authors present and discuss the coating of BC particles and associated optical and microphysical properties, but do not mention about the possible coating of BC on coarse mode dust particles. The latter situation is as important as the former one since it poses a great potential of changing the particles' radiative properties, thereby radiative forcing.

We thank the reviewer for raising this important aspect of BC research, which is that of the possible coating of BC on coarse mode dust particles. Dust is prevalent across northern India, especially in northern India, and plays an important role in altering

aerosol properties. In our study however, the SP2 is unable to account for coarse mode dust particles due to inherent limitations to the aerosol it can measure. Due to the inlet setup on the FAAM aircraft, and the size cut off in the SP2, our study only accounts for a BC characterisation (up to ∼500nm) not including dust. Work by Trembath et al (2012) examined the sampling inlet setup and constraints. They explain that submicron particle losses are considered negligibly for the operating altitudes that we considered during our study, but we are blind to the larger aerosol sizes involving refractory aerosol. In order to examine the coating of BC on coarse mode dust particles, we would have needed to operate and analyse filter measurements using other analysis techniques other than the SP2. Previous work by Liu et al (2018) explains this more in a study regarding aircraft measurements of dust aerosol. The SP2 has been found to measure hematite dust containing BC particles, peaking at size 650nm. This is above the maximum BC size that we measure as a part of this study, so this study is a characterisation of BC particles without this influence of dust. In our methodology it is stated, and now further clarified following these RC comments, that the possible coating of BC on coarse mode dust is not applicable to this study (page 6 line 31).

Second, the paper doesn't discuss, at least briefly, the similarities and/or discrepancies between the measured BC properties and those assumed in the climate-chemistry models for the study region. A general discussion highlighting the importance of new measurements given the current understanding of BC aerosols would greatly enhance the scientific value of the paper.

Most of previous studies analysing BC properties and characteristics over India have been based on analysis of satellite data, ground-based remote sensing or climate model simulations. However, there is currently large model-to-model variability suggestive of considerable uncertainty in model aerosol representation. We agree that our MS should discuss the discrepancies between measured and modelled BC properties over the study region, so we have now added this (page 13 line 31 onwards). We highlight that a lot of ambiguities arise from the uncertainty behind the vertical distribution of BC aerosol and its distribution relative to cloud cover. We explain that by providing high temporal and spatial resolution data for the pre-monsoon and monsoon seasons, future modelling work can implement the better-informed mass concentration and mixing state information that we provide.

The article is generally well-written, however, needs some attention to improve the presentation, e.g., long sentences, punctuations, and ambiguities. The content presented in the paper certainly fits into the scope of the ACP journal and can be published given above two major points are addressed. I am listing below some specific suggestions on the paper with this report. Page 2, Abstract, line 17: "..compared to that in the boundary layer" The text has been adapted to this change.

Page 2, Abstract: Define Indo-Gangetic Plain as IGP first in the abstract and use IGP for rest of the abstract to reduce the word count. Follow same terminology in Introduction and onwards. Complied with – we agree this is clearer.

Page 2, Abstract, line 29-31: The findings will also help constraint the regional aerosol models for a variety of applications such as space-based remote sensing, chemistry transport model, and air quality. Agreed – we have adapted the text to include this information as it is important.

Page 3, Introduction, line 5-7: IGP serves as a unique natural aerosol laboratory influenced by seasonal sources of aerosols. Authors may cite specific studies/papers out there that highlight seasonal loadings of dust and anthropogenic aerosols. Following the comments from RC2, the IGP material that was in the first paragraph has now been moved to paragraph 2. But regarding this comment from RC1, we have agreed and included citation of Vaishya et al (2018) and Gautam et al (2010) that highlight the seasonal loadings of dust and anthropogenic aerosol.

Page 3, Introduction, line 8-9: This is now a general understanding supported by numerous studies. Citing here the IPCC report or some high-impact factor on this subject would be more appropriate. Agreed – a citation for the IPCC report has now been

added.

Page 3, Introduction, line10: How about citing some benchmark papers of Tami Bond, who has done fundamental research on BC and its role in radiative heating. Agreed – this has now been included (referencing of Bond et al (2013) as well as from Marinoni et al (2010) and Bansal et al (2018)).

Page 3, Introduction, line13: I assume here that the central/southern Africa is the largest source of BC, and it is worth to mention it here with a corresponding citation. Yes, as highlighted in Bond et al. (2013) Figure 9 (page 5411 in article). This citation has now been added to clarify this.

Page 3, Introduction, line 16-17: What is the contribution of agricultural biomass burning in BC loadings over IGP? The region undergoes intensive crop residue burning in post-monsoon (paddy burning) and pre-monsoon (wheat burning) emitting substantial amounts of carbonaceous aerosols into the atmosphere. This has been outlined now in the manuscript: Indian BC emissions are overwhelmingly from low-efficiency combustion of domestic fuels (47%) followed by industrial emissions (22%), transportation (17%) and open burning (12%) which is dominated by crop residue burning (Paliwal et al., 2016).

Page 4, line 6: Define S2P here and then use abbreviation for the rest of the paper. Agreed – we have included the SP2 definition here.

Page 8, line 24: Some inconsistencies here. Please rewrite the sentence. Agreed – we have rewritten the sentence to remove the confusion over the regional pattern, removing inconsistencies.

Page 8, about coating thickness: Can the measurements and analysis shown here infer the properties of BC coating material? Mineral dust? Sulfate, Nitrate? Or humidification? Also, pre-monsoon period is also characterized with transported dust particles in the region. How about coating of dust with BC? Did the measurements/analysis show

BC coating coarse mode particles? This is important because studies have shown that the absorption properties of BC coating over dust are significantly different (more absorbing, lower SSA) than pure mineral dust (less absorbing, higher SSA). Unfortunately, we can't decipher the coating material. A previous paper by Brooks et al. (2019) presents what chemical species are seen during the same time/location as the BC presented here (SWAAMI flights), but the exact material involved in the coating is not possible with this methodology. Liu et al. (2018) describes this in more detail. They explain that a chemical approach to measure the chemical compositions of coatings associated with BC is achieved through a soot particle aerosol mass spectrometer (Aerodyne SP-AMS; Onasch et al., 2012). The SP2 instrument provides information on coating thicknesses through analysis of scattering and absorption signals but cannot provide properties of the BC coating material.

Page 8, line 29: I wouldn't locate Bhubaneswar in NE; it is actually towards South-East of Indian subcontinent. In terms of the IGP, we feel that yes Bhubaneswar would be more appropriately located as a South-East location. When comparing to other literature, we found Bhubaneswar was often referred to as on the eastern coast, eg: Sen et al. (2017), Kompalli et al. (2019). For continuity in our work against previous SWAAMI publication, we use NE India so that linkage can be made with a previous publication Brooks et al. (2019) where NE India is used for Bhubaneswar.

Page 8, line 30: Shouldn't it be Figure 4 and Figure 5? Agreed – this has been updated and corrected in the text.

Page 9, line 8-10: Confusing. Please consider restricting the sentences. This confusion has been cleared. We have altered the sentences to make this clear (please refer to the text in the manuscript).

Page 9: What is EAL? It isn't defined anywhere in the manuscript. The EAL definition has now been added, and it stands for Elevated Aerosol Layer. It refers to the aerosol that is present above the boundary mixed layer.
Page 11, Mass Absorption Coefficient, 1st paragraph: The possible reason for lower MAC over IGP and relative higher MAC in NW and NE isn't understood. In the NW boundary layer, MAC averaged 7.39 m2g-1 ($\sigma$ 0.36 m2g-1) compared to 6.76 m2g-1 ($\sigma$ 0.47 m2g-1) inside the IGP and 8.08 m2g-1 ($\sigma$ 0.33 m2g-1) over NE India. In our MS, we have stated that the MAC was lower in the IGP compared to NE and NW India potentially due to the proximity of measurements to the aerosol sources, with the MAC values high in the NE due to long-distance transport of BC in the pre-monsoonal wind flow travelling long distances through the IGP. Yes, the MAC values are lower in the IGP compared to outside, but when compared to the literature Laborde et al (2013) explains MAC values >8 refer to aged air masses and <7.5 refers to urban emissions. Over NE India, the air mass has undergone long range transport carrying IGP aerosol long distances, whereas in the central IGP and NW locations the measurements may have sampled aerosol closer to source.

Page 11, line 28-29: is it because of possibilities of elevated fine dust particles coated with BC? In our study we do not account for/measure dust. A possible reason for this increase in MAC values aloft compared to the BL may be due to the BC MAC dependency on mixing state (see Knox et al. (2009)). They explain that coatings that build up on BC can act as a lens, focusing light into the BC core, increasing absorption per mass of BC. Also, mixing state is a function of particle age amongst other factors. Aloft over the IGP the BC present will have undergone atmospheric aging and experienced long-range transport, therefore providing a potential reason for the larger MAC values. This information has now been added to the MS. 

acp-2019-505-RC2 Reviewer Comments to Author(s): This manuscript presents the physical and optical properties of black carbon (BC) over northern India using a single particle soot photometer (SP2). The study mainly focusses on the Indo-Gangetic Plain (IGP) during the pre-monsoon and monsoon seasons. The ms brings detailed information about spatial and vertical distribution of BC aerosols that can help to improve understanding of the role of BC in radiative forcing and climate models. The

work presented in the manuscript well fits into the scope of the journal. Overall, the article is well-structured; however, needs some attention to improve the value/quality of the manuscript. We thank the Reviewer for their encouraging view of our work and we hope that the suggested alterations have been carried out fully and improve the manuscript.

General comments: The authors need to discuss more details about the coating of black carbon, dust particles. We thank the reviewer for raising this important aspect of BC research, which is that of the possible coating of BC on coarse mode dust particles. Dust is prevalent across northern India and plays an important role in altering aerosol properties. However, in our study we concentrate solely on BC. In summary, the SP2 is unable to account for coarse mode dust particles due to inherent limitations to the aerosol it can measure. Due to the inlet setup on the FAAM aircraft, and the size cut off in the SP2, our study only accounts for a BC characterisation (up to ∼500nm) not including dust. Please refer to the response to RC1 previous for further explanation about coating of black carbon, dust particles.

How the current work is relevant and implements to climate models over the study area?

We have now added a discussion into how our work is relevant and addresses current ambiguities between BC measurements and modelling work, as explained for the RC1 comment (see page 13 line 31 onwards).

Also, authors need to use more abbreviations like IGP, BC, SP2 throughout the manuscript.

We have now made sure that abbreviations are used throughout the manuscript in places where this was lacking before.

I am listing below some specific suggestions that can help to improve the quality of the manuscript.

[Figure]

Abstract

The abstract need to revise as the current description is very general, some information possible to move (line 2-10) in the introduction section.

Agreed – we have removed some of the information from the abstract in lined 2-10 as it is mentioned in the introduction already.

Use abbreviation of Indo-Gangetic Plain (IGP) as it repeated in several places (line 11, 14, 18, 21, 23, 24 and 27).

Complied with – this has been changed to use IGP.

Please use shorter sentences instead of a long sentence (line 19-24).

In its current state, line 19-24 is split into two sentences. We feel this is most appropriate to avoid over complicating the text and splitting into too many sentences, in this instance.

Introduction Page 3: line 2, need references to support the sentence

We have used the reference of Schnell et al. (2018) to support this sentence.

Page 3: line 13-24, the second paragraph need to revise with good connection with earlier sentences. Also possible to discuss studies which are mainly focusing on BC source apportionment over South Asia.

Complied with – we have moved the IGP information into the second paragraph in this section so the text does not switch between location commentary in the 2 paragraphs, so it is more connected now. For discussing studies focusing on BC source apportionment over South Asia, citations have been added for this (Moorthy et al, 2013; Paliwal et al, 2016).

Page 4: line 19-21, possible to combine these two sentences

Complied with – we agree this flows better in the text to combine, so have done so.

Methodology This section is too descriptive so better to move some information is supplement information, this will make more space for results and discussion.

Regarding this suggestion, we have decided to not move to a supplementary information section. This is mainly due to the equations and background information, as for other similar papers that include SP2 black carbon analysis (e.g. Liu et al., 2018; 2019), usually existing in the methods section. In order for the reader to be able to understand how the coating thickness etc are calculated, we feel it is most appropriate to leave this material in the main text.

Results Page 7, line 2, Please start this important section with topic sentences rather than measurements

Complied with – we have outlined the topic material at the start now, guiding the reader through the material and sections of such material, before outlining the summary measurement analysis that provides the context for the BC results presented next.

Page 7, line 18-19, use IGP abbreviation

We agree – the IGP abbreviation has now been used.

Page 7, line 15-18, Please explain this in more details

Regarding detail here, this was added: sulphate aerosol dominating outside the IGP compared to organic aerosol inside the IGP. More detail is provided in the citation provided, this paragraph is to give context to the aerosol vertical structure and the changes that occur as the monsoon progresses over northern India.

Page 8, 1-2, Need to add more details to support this statement

Agreed – we have now added in more detail surrounding the reasons for larger coating thicknesses in the NE region. A reason for these large coating thicknesses may be due to the air mass arriving in the north-east having been strongly influenced by long-range transport that has undergone significant aging mechanisms, as well as entrainment of

solid fuel aerosol particles as the air mass travels across the IGP.

Conclusions

Please revise this full section with more specific outcomes.

We have revised the conclusion section to comply with this suggestion of covering the more specific outcomes. Throughout, we have added in specific information regarding the main findings, providing the values behind the findings. Also, coverage has been provided on the consistency between aerosol source analysis and the black carbon characteristics measured across the IGP in particular in order to highlight the specific importance and consistency of our results compared to the literature.

Author contributions

Please write a full description of SWAAMI project.

We agree this is useful for the wider scientific audience, so we have included a fuller description of the SWAAMI project here, but more information can be found in the earlier Methodology section (page 4 line 26 onwards) as well as in earlier publications from the SWAAMI project (e.g. Brooks et al., 2019 and Vaishya et al., 2019).

  References

Bansal, O., Singh, A., and Singh, D.: Characteristics of Black Carbon aerosols over Patiala Northwestern part of the IGP: Source apportionment using cluster and CWT analysis. Atmospheric Pollution Research, 10(1), 244–256. http://doi.org/10.1016/j.apr.2018.08.001, 2018. Bond, T.C., Doherty, S.J., Fahey, D.W., Forster, P.M., Berntsen, T., DeAngelo, B.J., Flanner, M.G., Ghan, S., Kärcher, B., Koch, D. and Kinne, S.: Bounding the role of black carbon in the climate system: A scientific assessment. Journal of Geophysical Research: Atmospheres, 118(11), pp.5380-5552, 2013. Brooks, J., Allan, J. D., Williams, P. I., Liu, D., Fox, C., Haywood, J., Langridge, J. M., Highwood, E. J., Kompalli, S. K., O'Sullivan, D., Babu, S. S., Satheesh, S. K., Turner, A. G., and Coe, H.: Vertical and horizontal distribution

of submicron aerosol chemical composition and physical characteristics across northern India during pre-monsoon and monsoon seasons, Atmos. Chem. Phys., 19, 5615–5634, https://doi.org/10.5194/acp-19-5615-2019, 2019. Gautam, R., Hsu, N. C., and Lau, K. M.: Premonsoon aerosol characterization and radiative effects over the Indo-Gangetic plains: Implications for regional climate warming, J. Geophys. Res.-Atmos., 115, D17208, https://doi.org/10.1029/2010JD013819, 2010.  Knox, A., Evans, G.J., Brook, J.R., Yao, X., Jeong, C.H., Godri, K.J., Sabaliauskas, K. and Slowik, J.G.: Mass absorption cross-section of ambient black carbon aerosol in relation to chemical age. Aerosol science and technology, 43(6), pp.522-532, 2009. Kompalli, S. K., Suresh Babu, S. N., Satheesh, S. K., Krishna Moorthy, K., Das, T., Boopathy, R., Liu, D., Darbyshire, E., Allan, J., Brooks, J., Flynn, M., and Coe, H.: Seasonal contrast in size distributions and mixing state of black carbon and its association with PM1.0 chemical composition from the eastern coast of India, Atmos. Chem. Phys. Discuss., https://doi.org/10.5194/acp-2019-376, in review, 2019. Liu, D., Taylor, J. W., Crosier, J., Marsden, N., Bower, K. N., Lloyd, G., Ryder, C. L., Brooke, J. K., Cotton, R., Marenco, F., Blyth, A., Cui, Z., Estelles, V., Gallagher, M., Coe, H., and Choularton, T. W.: Aircraft and ground measurements of dust aerosols over the west African coast in summer 2015 during ICE-D and AER-D, Atmos. Chem. Phys., 18, 3817–3838, https://doi.org/10.5194/acp-18-3817-2018, 2018. Liu, D., Joshi, R., Wang, J., Yu, C., Allan, J. D., Coe, H., Flynn, M. J., Xie, C., Lee, J., Squires, F., Kotthaus, S., Grimmond, S., Ge, X., Sun, Y., and Fu, P.: Contrasting physical properties of black carbon in urban Beijing between winter and summer, Atmos. Chem. Phys., 19, 6749–6769, https://doi.org/10.5194/acp-19-6749-2019, 2019. Marinoni, A., Cristofanelli, P., Laj, P., Duchi, R., Calzolari, F., Decesari, S., Sellegri, K., Vuillermoz, E., Verza, G.P., Villani, P. and Bonasoni, P.: Aerosol mass and black carbon concentrations, a two year record at NCO-P (5079 m, Southern Himalayas). Atmospheric Chemistry and Physics, 10(17), pp.8551-8562, 2010. Moorthy, K.K., Beegum, S.N., Srivastava, N., Satheesh, S.K., Chin, M., Blond, N., Babu, S.S. and Singh, S.: Performance evaluation of chemistry transport models over India.

Atmospheric environment, 71, pp.210-225, 2013. Onasch, T., Trimborn, A., Fortner, E., Jayne, J., Kok, G., Williams, L., Davidovits, P., and Worsnop, D.: Soot particle aerosol mass spectrometer: development, validation, and initial application, Aerosol Sci. Technol., 46, 804–817, 2012 Paliwal, U., Sharma, M., and Burkhart, J. F.: Monthly and spatially resolved black carbon emission inventory of India: uncertainty analysis, Atmos. Chem. Phys., 16, 12457-12476, https://doi.org/10.5194/acp-16-12457-2016, 2016. Sen, A., Abdelmaksoud, A.S., Ahammed, Y.N., Banerjee, T., Bhat, M.A., Chatterjee, A., Choudhuri, A.K., Das, T., Dhir, A., Dhyani, P.P. and Gadi, R.: Variations in particulate matter over Indo-Gangetic Plains and Indo-Himalayan Range during four field campaigns in winter monsoon and summer monsoon: role of pollution pathways. Atmospheric environment, 154, pp.200-224, 2017. Trembath, J., Bart, M., and Brooke, J.: Efficiencies of modified rosemount housings for sampling aerosol on a fast atmospheric research aircraft, availabe at: https://old.faam.ac.uk/index.php/faam-documents/science-instruments/1673-inlet-efficiency/file, FAAM Technical Note, 27 pp., 2012. Vaishya, A., Babu, S. N. S., Jayachandran, V., Gogoi, M. M., Lakshmi, N. B., Moorthy, K. K., and Satheesh, S. K.: Large contrast in the vertical distribution of aerosol optical properties and radiative effects across the Indo-Gangetic Plain during the SWAAMI-RAWEX campaign, Atmos. Chem. Phys., 18, 17669–17685, https://doi.org/10.5194/acp-18-17669-2018, 2018. 

Please also note the supplement to this comment:
https://www.atmos-chem-phys-discuss.net/acp-2019-505/acp-2019-505-AC1-supplement.pdf
* * *